# Toward an Early Diagnosis for Alzheimer’s Disease Based on the Perinuclear Localization of the ATM Protein

**DOI:** 10.3390/cells12131747

**Published:** 2023-06-29

**Authors:** Elise Berthel, Laurent Pujo-Menjouet, Eymeric Le Reun, Laurène Sonzogni, Joëlle Al-Choboq, Abdennasser Chekroun, Adeline Granzotto, Clément Devic, Mélanie L. Ferlazzo, Sandrine Pereira, Michel Bourguignon, Nicolas Foray

**Affiliations:** 1Institut National de la Santé et de la Recherche Médicale, U1296 Research Unit «Radiation: Defense, Health, Environment», Centre Léon-Bérard, 28 Rue Laennec, 69008 Lyon, France; eberthel@neolys-diagnostics.fr (E.B.); eymeric.le-reun@inserm.fr (E.L.R.); laurene.sonzogni@inserm.fr (L.S.); joelle.al-choboq@inserm.fr (J.A.-C.); adeline.granzotto@inserm.fr (A.G.); ferlazzo@ansto.au (M.L.F.); michel.bourguignon@inserm.fr (M.B.); 2NEOLYS Diagnostics, 7 Allée de l’Europe, 67960 Entzheim, France; spereira@neolys-diagnostics.fr; 3Université Claude-Bernard Lyon 1, CNRS UMR5208, INRIA, Institut Camille-Jordan, 21 Avenue Claude Bernard, 69603 Villeurbanne, France; pujo@maths.univ-lyon1.fr (L.P.-M.); checkroun@maths.univ-lyon1.fr (A.C.); 4Université Paris-Saclay, 78035 Versailles, France

**Keywords:** Alzheimer’s disease, ATM, APOE, aging, fibroblasts, immunofluorescence, DNA double-strand breaks, irradiation

## Abstract

Alzheimer’s disease (AD) is the most common neurodegenerative dementia, for which the molecular origins, genetic predisposition and therapeutic approach are still debated. In the 1980s, cells from AD patients were reported to be sensitive to ionizing radiation. In order to examine the molecular basis of this radiosensitivity, the ATM-dependent DNA double-strand breaks (DSB) signaling and repair were investigated by applying an approach based on the radiation-induced ataxia telangiectasia-mutated (ATM) protein nucleoshuttling (RIANS) model. Early after irradiation, all ten AD fibroblast cell lines tested showed impaired DSB recognition and delayed RIANS. AD fibroblasts specifically showed spontaneous perinuclear localization of phosphorylated ATM (pATM) forms. To our knowledge, such observation has never been reported before, and by considering the role of the ATM kinase in the stress response, it may introduce a novel interpretation of accelerated aging. Our data and a mathematical approach through a brand-new model suggest that, in response to a progressive and cumulative stress, cytoplasmic ATM monomers phosphorylate the APOE protein (pAPOE) close to the nuclear membrane and aggregate around the nucleus, preventing their entry in the nucleus and thus the recognition and repair of spontaneous DSB, which contributes to the aging process. Our findings suggest that pATM and/or pAPOE may serve as biomarkers for an early reliable diagnosis of AD on any fibroblast sample.

## 1. Introduction

Alzheimer’s disease (AD) is a progressive and devastating neurodegenerative disorder with a poor prognosis, and for which no efficient curative treatment is currently available [1,2]. AD has been shown to result in specific neurobiological events, such as extracellular deposits of β-amyloid (Aß) plaques [3] and intracellular neurofibrillary tangles consisting in hyper-phosphorylated aggregates of the microtubule-associated tau protein [4]. Interestingly, cellular radiosensitivity was also reported in lymphocytes derived from AD patients [5,6,7]. However, to date, a mechanistic model linking the abnormal response to ionizing radiation (IR) and the accelerated neurodegeneration/aging observed in AD patients in response to oxidative stress, whether endogenous or exogenous, is still needed.

In 2016, from one of the largest collections of cutaneous fibroblasts derived from patients suffering from post-radiotherapy radiosensitivity, a unified mechanistic model of individual responses to IR was proposed [8,9]. This model was based on the radiation-induced (RI) nucleoshuttling of the ataxia telangiectasia-mutated (ATM) protein (RIANS) required for signaling and repair of DNA double-strand breaks (DSB), the key DNA damage in response to IR. The RIANS model has been shown to be the most reliable predictor of radiosensitivity [8,10,11,12,13,14], and to provide a molecular explanation of some low-dose-specific radiobiological phenomena [9,15,16,17]. It was also validated in non-radiative oxidative stress, such as contamination with metals and pesticides [18,19]. In the frame of the RIANS model, any oxidative stress induces DNA breaks and the monomerization of ATM dimers. Cytoplasmic ATM monomers diffuse thereafter in the nucleus and trigger the DSB recognition and repair via the non-homologous end-joining (NHEJ), the predominant DSB signaling and repair pathway in quiescent human cells [8,9]. Any delayed RIANS was shown to be associated with radiosensitivity (RI cell lethality), radio-susceptibility (RI cancers) and/or radio-degeneration (RI accelerated aging) [17,20]. Delayed RIANS is generally caused by the sequestration of cytoplasmic ATM monomers by overexpressed ATM substrate proteins, called X-proteins [17,21]. A number of X-proteins have already been identified in neurodegenerative/aging or cancer-prone diseases associated with radiosensitivity, therefore validating the RIANS model. This is notably the case of Huntington’s disease [22], tuberous sclerosis complex [23], Xeroderma Pigmentosum D [24], retinoblastoma [25], neurofibromatosis type I [26] and Rothmund–Thomson syndromes [27]. It is noteworthy that the occurrence of cytoplasmic forms of ATM, the basis of the RIANS model, has been initially supported by several other research groups [28,29,30,31,32].

Here, the major steps of the ATM-dependent RI DSB signaling and repair pathway were investigated in cutaneous fibroblasts derived from 10 AD patients. The AD data were compared to those obtained from fibroblasts derived from 5 apparently healthy and radioresistant donors and 11 patients suffering from cancer or neurodegenerative diseases. During our investigations, a spontaneous perinuclear localization of the ATM protein was observed specifically in the AD fibroblasts, which prompted us to propose a mechanistic interpretation of AD deriving from the RIANS model and consolidated by a biomathematical approach.

## 2. Materials and Methods

### 2.1. Cell Culture

All the experiments were performed with human untransformed cutaneous fibroblasts that were routinely cultured as monolayers with Gibco modified Eagle’s minimum medium (DMEM) (Thermo Fisher, Waltham, MA, USA), supplemented with 20% fetal calf serum, penicillin and streptomycin (Thermo Fisher). All the experiments were performed with cells in the plateau phase of growth to avoid any cell cycle effect [33]. The genetic and cellular features of the cutaneous fibroblasts tested are detailed in Table 1. The 10 untransformed AD fibroblasts were purchased from the Coriell Institute for Medical Research (Camden, NJ, USA). All the non-AD fibroblasts used in this study were purchased either from the European Collection of Authenticated Cell Cultures (ECACC, UK Health Security Agency, Salisbury, UK), the American Type Culture Collection (ATCC, Manassas, VI, USA), the Coriell Institute or from the COPERNIC collection of our lab that gathers primary fibroblasts of different origin and radiosensitivity. The COPERNIC collection was approved by a regional ethical committee, declared under the numbers DC2011-1437 and 2021-3957 to the Ministry of Research as required by the French regulations, and was already described elsewhere [8]. It obeys the French regulations about the anonymous sampling and the informed consent. Cell authentication and culture quality control were insured by commercial repositories. With regard to the cell lines isolated from the lab (COPERNIC collection), the respect of the good practices of the laboratory (GPL) was included in the declaration of the cell line collection to the French Ministry of Research.

### 2.2. Treatment with Zoledronate and Pravastatine (ZOPRA)

The combination of zoledronate and pravastatin was called ZOPRA treatment, as described elsewhere [34], and it was applied to more than ten different genetic diseases associated with a delayed RIANS [35]. Briefly, cells were incubated with 1 µM of pravastatin (Sigma-Aldrich, Saint-Quentin-Fallavier, France) in phosphatase-buffered saline solution (PBS) for 24 h at 37 °C. Thereafter, 1 µM of zoledronate (Sigma-Aldrich) in PBS was added into the culture medium and cells were incubated for 12 h at 37 °C. The culture medium was renewed immediately before irradiation.

### 2.3. Irradiation

All the irradiations were performed on a 6 MeV X-ray clinical irradiator (SL 15 Phillips) at the Anti-Cancer Centre Léon-Bérard (Lyon, France), at a dose of 2 Gy with a dose rate of 6 Gy min^−1^. Dosimetry was certified by the Radiophysics Department of Centre Léon-Bérard.

### 2.4. Clonogenic Cell Survival

The intrinsic cellular radiosensitivity was quantified by using the clonogenic cell survival assay, for which the protocol was detailed elsewhere [36]. Each survival data point is the mean of at least 3 replicates with 3 different dilutions tested per dose. The number of cells seeded ranged from 200 to 10,000 cells per 6 cm-diameter Petri dish [36]. The intrinsic radiosensitivity was quantified by the surviving fraction at 2 Gy (SF2) [9].

### 2.5. Phosphospecific APOE Antibodies

Two custom rabbit polyclonal phosphospecific *APOE* antibodies were developed against pAPOE^ser72^ and pAPOE^thr75^ residues at our request by the CliniSciences company (Nanterre, France) by following standard procedures developed by the manufacturer.

### 2.6. Immunofluorescence

Immunofluorescence and foci scoring procedures were described elsewhere [8,24,37]. Briefly, the anti-*γ-H2AX^ser139^* antibody (clone JBW301; Merck, Millipore, Darmstadt, Germany) was applied at 1:800. Anti-*pATM^ser1981^* (clone 10H11.E12; Millipore, Germany) and anti-*MRE11* (#56211; Abcys, Paris, France) were used at 1:100. Cells were counterstained with 4’,6-diamidino-2-phenylindole, dihydrochloride (DAPI), which also permitted to score the micronuclei in the same conditions. The foci scoring procedure applied here has received the certification agreement of CE mark and ISO-13485 quality management norms and developed some features protected in the frame of the patents FR3017625 A1 and FR3045071 A1. The immunofluorescence data resulted in at least 3 independent replicates with 100 nuclei analyzed per experiment and per biomarker.

### 2.7. Micronuclei Assay

During each immunofluorescence experiment, the DAPI counterstaining permitted to quantify the micronuclei at magnification ×100 [14]. The micronuclei data resulted in at least 3 independent replicates with 100 nuclei analyzed per experiment and per biomarker.

### 2.8. Immunoprecipitation and Immunoblotting

Total extracts were obtained from 1 to 2 × 10^6^ cells collected by scraping and rinsed by centrifugation with cold phosphate-buffered saline (1500 rpm, 200× *g*, 4 °C, 5 min), then lysed in lysis buffer A (50 mM Tris-HCl, pH 8, 150 mM NaCl, 2 mM EDTA, 0.2% NP40 nonidet and 10% glycerol) supplemented with protease and phosphatase inhibitors (#78442, Thermo Fisher, Waltham, MA, USA) and maintained in 4 °C for 15 min. Total extracts were finally obtained in the supernatant after centrifugation (1500 rpm, 200× *g*, 4 °C, 5 min). Cytoplasmic extracts were obtained from 1 to 2 × 10^6^ rinsed cells using the buffer (10 mM Hepes pH 7.9, 1.5 mM MgCl_2_, 10 mM KCL, 2 mM ethylenediaminetetraacetic acid (EDTA) pH 8, 0.5 mM dithiothreitol (DTT), 0.2% Nonidet NP40, H_2_O), supplemented with protease and phosphatase inhibitors (#78442, Thermo Fisher, Waltham, MA, USA) and maintained in 4 °C for 15 min. Cytoplasmic extracts were finally obtained in the supernatant after centrifugation (5000 rpm, 800× *g*, 4 °C, 5 min). After this procedure, nuclear extracts were obtained from the residual pellet lysed in the following buffer (10 mM Hepes pH 7.9, 1.5 mM MgCl_2_, 450 mM KCL, 2 mM EDTA, 1 mM ethyl-glycol-bis(β-aminoethylether-tetraacetic acid) (EGTA) pH 8, 0.5 mM dithiothreitol (DTT), 0.25% glycerol, H_2_O), supplemented with protease and phosphatase inhibitors (#78442, Thermo Fisher, Waltham, MA, USA) applied for 1 h, at 4 °C. Nuclear extracts were finally obtained from the supernatant after centrifugation (5000 rpm, 800× *g*, 4 °C, 5 min). Protein concentrations were determined by the Bradford procedure (Bio-Rad Laboratories, Hercules, CA, USA). For immunoprecipitations, lysates containing 1.5 mg of protein were pre-cleared by stirring with 40 μL of Protein G–Sepharose beads (GE Healthcare, Chicago, IL, USA) for 1 h, at 4 °C. After centrifugation, immunoprecipitation was performed with the pre-cleared lysate and 2 μg of antibody for 3 h, at 4 °C. A 40 μL volume of Protein G–Sepharose was added and incubated for 30 min at 4 °C. After centrifugation, beads were washed three times with lysis buffer A and proteins were eluted by heating at 95 °C for 5 min in SDS loading buffer and 100 mM of dithiothreitol (DTT) (Sigma-Aldrich). Proteins were subjected to SDS-PAGE and blotted onto PVDF membranes (Immobilon-P, Millipore, Burlington, MA, USA). Membranes were blocked in Tris-buffered saline (TBS) solution containing 0.05% Tween 20 and 5% (*w*/*v*) non-fat dried skimmed milk powder and incubated with primary antibodies. Horseradish peroxidase-conjugated secondary antibodies (Jackson ImmunoResearch-Europe, Ely, UK) were used for the detection of immunoreactive proteins via the ECL kit (Thermo Scientific, Waltham, MA, USA).

### 2.9. In Situ Proximity Ligation Assay (PLA)

The PLA technology allows the in situ detection of endogenous protein interactions by immunofluorescence microscopy [38]. This technique consists in recognition of two proteins of interest using secondary antibodies conjugated with DNA strands. When in close proximity (≤10 nm), these strands form a template for rolling-circle amplification using fluorescent oligonucleotides. The interaction between the two proteins of interest is visible as a distinct fluorescent spot. Cells were fixed with 4% (*w*/*v*) paraformaldehyde for 15 min at room temperature and were permeabilized in 0.5% Triton X-100 solution (Sigma-Aldrich) for 3 min at 4 °C. Cells were then blocked for 2 h at room temperature using 30 µL of blocking solution from the Duolink^TM^ In Situ Orange Starter Kit Mouse/Rabbit (#DUO92102, Sigma-Aldrich) per coverslip. Mixtures of two primary antibodies’ incubations were performed for 1 h at 37 °C (one raised in rabbit, the other one in mouse). The following antibodies were diluted in the Duolink antibody diluent 1X (#DUO82008-Sigma) at a ratio of 1:100: mouse monoclonal anti-*ATM* antibody (#2C1 (1A1); Abcam, Cambridge UK) and rabbit polyclonal anti-*APOE* antibody (#SAB2701971; Sigma-Aldrich). PLA probes (Duolink PLA Probe anti-mouse MINUS and Duolink PLA Probe anti-rabbit PLUS, Olink AB) were diluted the same as all Duolink reagents. After 1 h of incubation at 37 °C with the PLA probes, cells were washed in buffer A (10 mM Tris/HCl, pH 7.4, 150 mM NaCl and 0.05% Tween 20) and treated with the Duolink In Situ Detection Reagents Orange (Olink AB, Sigma-Aldrich). Cells were incubated with the ligation solution for 30 min at 37 °C in the humidified chamber. After washing in buffer A, cells were incubated with the amplification solution for 100 min at 37 °C in darkness in the humidified chamber. Cells were then washed with wash buffer B (DUO82048-1EA, Sigma-Aldrich), followed by a quick wash with 1/100 wash buffer B. Samples were mounted with Duolink In Situ Mounting Medium with DAPI (DUO82040; Sigma-Aldrich). Samples were analyzed under fluorescent microscopy. Fluorescent signals were viewed under an Olympus BX51 microscope (×100 objective). Analysis and quantification of these samples were performed using the ImageJ 1.52g software. The PLA data resulted in at least 3 independent replicates with 100 nuclei analyzed per experiment and per biomarker couple.

### 2.10. Statistical Analysis

All data were obtained with the numbers of independent experiments indicated and each value was expressed as the mean and standard error of the mean (Poisson’s law). The statistical analyses were performed either by PRISM software version 9.5.1 (GraphPad Software, San Diego, CA, USA) or by using Kaleidagraph version 4.5.4 (Synergy Software, Reading, PA, USA). One-way ANOVA tests were performed for point-to-point analysis.

### 2.11. Mathematical Modeling

A brand-new mathematical model was designed to unify all four patterns (no pATM crown, no pATM crown with foci, thin pATM crown with foci, thick pATM crown without foci) in a single system of nonlinear differential equations, and based on the description of protein interactions, in six compartments: the ATM dimers in cytoplasm, the ATM monomers in cytoplasm, the ATM dimers inside the perinuclear crown, the APOE-ATM complexes, the APOE proteins, and finally, the ATM monomers in the nucleus. Each interaction from one compartment to another was represented by a term of the equation (interaction, nucleus passage, monomerization, dimerization, complex formation) with specific kinetics. Some of these kinetic parameters were protein-dependent (such as k4 depending on AM (that is the APOE-ATM complexes with saturation) and k5 depending inversely on D_A_ (that is the ATM dimers in the perinuclear crowns)). The simulation results led to one of the four patterns regarding the values of the parameters and/or initial conditions.

## 3. Results

### 3.1. AD Fibroblasts Show Radiosensitivity Associated with a Delayed RIANS

Cellular radiosensitivity is currently quantified by the clonogenic cell survival fraction at 2 Gy (SF2) [39]. While the average SF2 values from the radioresistant controls (65.6 ± 1.8%) were in good agreement with the literature [39,40,41], the AD fibroblasts tested showed significantly lower SF2 than controls (average SF2 = 40.2 ± 6.6%; *p* < 0.01) (Figure 1A). The SF2 values of AD cells suggested a moderate but significant radiosensitivity, quantitatively similar to that of Werner’s syndrome or Huntington’s disease, but significantly lower than that observed in ataxia telangiectasia (AT), the human syndrome associated with the highest radiosensitivity (average SF2 = 1.5 ± 0.3%; *p* < 0.001) [21,39,40,41] (Figure 1A). The SF2 values of AD fibroblasts were found in agreement with those from AD lymphocytes published in the 1980s [5,6,7].

Micronuclei are cytogenetic endpoints resulting from the propagation of unrepaired DSB to G2/M phase and their transformation as chromosome fragments [14]. The yield of micronuclei has been shown to be quantitatively correlated with cellular radiosensitivity [14,20]. In the 10 AD fibroblasts tested, the yields of spontaneous and RI micronuclei were significantly higher than those of radioresistant controls (*p* < 0.01 and *p* < 0.05, respectively) (Figure 1B). SF2 and micronuclei data from AD fibroblasts were found consistent with correlations with 200 different human cell lines, showing a large spectrum of radiosensitivity [14] (Appendix A).

Since the ATM phosphorylation of the X variant of the H2A histone (γH2AX) was demonstrated to form nuclear foci at the RI DSB sites recognized by NHEJ [42], anti-*γH2AX* immunofluorescence was applied to AD fibroblasts. While no difference was observed at 4 and 24 h post-irradiation, the number of γH2AX foci assessed 10 min and 1 h post-irradiation was found systematically lower in the AD fibroblasts than in the radioresistant controls (*p* < 0.01 and *p* < 0.05, respectively) (Figure 2). These findings were found similar to those obtained from cells deriving from syndromes associated with moderate radiosensitivity, cancer proneness or accelerated aging [22,23,24,25,26]. It is noteworthy that the γH2AX foci were either absent or impaired in AT cells, in agreement with the literature [8,27]. Again, γH2AX data from AD data obeyed a correlation between γH2AX and SF2 data established with 200 human fibroblast cell lines [14] (Appendix A).

In the frame of the RIANS model, the above data suggest a lack of ATM-dependent RI DSB recognition by NHEJ [8]. At the end of the repair process, trans-autophosphorylated ATM (pATM) monomers formed ATM dimers that were revealed by the formation of pATM foci in the nucleus and by a broader staining in the cytoplasm [17]. It is noted that pATM-positive signals indicate the presence of ATM dimers, while the anti-*ATM* antibody cannot discriminate ATM monomers and dimers [17,32]. By applying anti-*pATM* immunofluorescence, the number of RI pATM foci in AD fibroblasts at 10 min post-irradiation appeared significantly lower than in the radioresistant controls (*p* < 0.01) (Figure 3). Such observation was consolidated by pATM immunoblots of nuclear extracts that showed a lower increase of the nuclear pATM molecules at 10 min post-irradiation in AD fibroblasts than in the radioresistant controls (Figure 3). It is noteworthy that qPCR analysis of ATM synthesis did not reveal any difference between the radioresistant controls and the AD fibroblasts, suggesting that the RIANS is independent of the ATM protein synthesis. Besides, the pATM data observed at 10 min post-irradiation were unlikely due to any change of ATM expression that may generally occur some hours post-irradiation. Again, these findings were consistent with the pATM data observed in syndromes associated with moderate radiosensitivity, cancer proneness or accelerated aging [22,23,24,25,26]. Lastly, it must be stressed that AD data also obeyed a correlation between pATM and SF2 data that has been established with 200 other human fibroblast cell lines [14] (Appendix A).

In our previous reports, a delayed RIANS was generally associated with abnormal radiation-induced re-localization of nuclear foci formed by the MRE11 nuclease. Particularly, cells from aging syndromes were found characterized by a delayed production of MRE11 foci, while cells from cancer syndromes generally showed early MRE11 foci [21]. By applying anti-*MRE11* immunofluorescence, all the AD fibroblasts tested elicited a number of nuclear MRE11 foci at 24 h post-irradiation, significantly higher than that of the controls (*p* < 0.01), again supporting the relationship between late MRE11 foci and aging (Figure 4).

Altogether, our findings suggested that AD is associated with significant but moderate cellular radiosensitivity with impaired yields of micronuclei, γH2AX, pATM and MRE11 foci, consistent with a delayed RIANS. It must be stressed here that the radiobiological features of the 10 AD fibroblast cell lines tested were not found to be dependent on age, gender or the genotype of the corresponding donors (Table 1 and Appendix A). Furthermore, all the relationships between the different RIANS biomarkers tested obeyed inter-correlations recently established with 200 fibroblast cell lines, showing a large spectrum of radiosensitivity [14] (see Appendix A), which suggests a strong quantitative relevance between the data assessed in AD cells.

### 3.2. Combination of Zoledronate and Pravastatin May Partially Protect Some AD Fibroblasts from IR

In the frame of the RIANS model, it was shown that bringing more ATM monomers into the nucleus significantly contributed to enhancing the radioresistance [17,35]. A radioprotective effect with the combination of bisphosphonates and statins, particularly with zoledronate and pravastatin (ZOPRA), was reported: the ZOPRA treatment accelerated the diffusion of the ATM monomers across the nuclear membrane in a number of genetic diseases [22,23,24,26,35]. Surprisingly, when applied to the 10 AD fibroblast cell lines, the ZOPRA treatment did not significantly change the yield of RI micronuclei (Figure 5A). It contributed to increase the number of γH2AX (Figure 5B) and pATM foci assessed 10 min post-irradiation, but this trend was not significant for all the cell lines tested (Figure 5C). Regarding the MRE11 foci, the late MRE11 foci disappeared in only 4 of the 10 AD fibroblast cell lines (Figure 5D). Altogether, these findings suggest that the ZOPRA treatment may protect, at least partially, some AD fibroblasts from IR, while it has been applied more successfully in cancer syndromes with similar radiobiological features as AD cells [35]. It must be stressed that the ZOPRA treatment is not efficient in protecting the *ATM*-mutated cells since it cannot overcome the loss of the ATM protein function [35]. Besides, applying ATM inhibitors in our conditions would not help to understand the specificities of AD cells since the activity of ATM kinase is required for the formation of ATM dimers, whether cytoplasmic or nuclear. The partial action of ZOPRA treatment in AD cells prompted us to hypothesize that ZOPRA molecules’ diffusion in the nucleus was specifically prevented in AD cells, and we further investigated the cellular features of AD cells.

### 3.3. AD Fibroblasts Spontaneously Show Specific Abnormal Perinuclear Localization of the ATM Protein

During the immunofluorescence experiments described above, a strong perinuclear pATM staining was noticed in non-irradiated AD fibroblasts, suggesting an abnormally high concentration of ATM dimers around the nuclear membrane. The percentage of cells with perinuclear pATM crowns varied from 7 to 44 at the lowest passages tested, among the AD cell lines tested (Figure 6A,B). No correlation was found between the percentage of cells with perinuclear pATM crowns and the age at the skin sampling gender or known mutation status of the AD donors, consistent with the fact that these last three endpoints are not necessarily linked to the AD progress (Appendix A). In contrast, the percentage of cells with perinuclear pATM crowns in AD cells progressively increased as a function of the cell culture passage and reached a plateau, suggesting a sigmoidal function of the passage with a threshold (ranging from passage 7 to 10) and a plateau (ranging from 20% to 40% of cells with perinuclear pATM crowns) (Figure 6C).

The perinuclear pATM crowns were totally absent in radioresistant controls cultured in the same passage range, whether irradiated or not (Figure 6B). Again, in the same passage range, no perinuclear pATM crown was observed in the non-AD fibroblasts derived from the other genetic syndromes described in Table 1, regardless of their radiosensitivity and whether they were irradiated or not (Figure 6B). It must be stressed that this statement is particularly true for fibroblasts from Huntington’s disease (HD), Hutchinson–Gilford progeroid syndrome (HGPS), Xeroderma Pigmentosum D (XPD), Werner’s syndrome (WS) and tuberous sclerosis complex syndrome (TSC), suggesting a strong specificity of this molecular feature for AD among the neurodegenerative or aging diseases tested (Figure 6B). Lastly, no perinuclear pATM crown was observed in the 200 radiosensitive skin fibroblasts of the COPERNIC collection derived from non-AD cancer patients treated by radiotherapy [8].

In each of the non-irradiated AD fibroblast cell lines tested, we identified four different patterns of pATM staining with different relative proportions, supporting a great heterogeneity in the AD fibroblast populations, but also an ordered temporal progression among the different patterns: (1) Cytoplasmic pATM broad staining without nuclear pATM foci, reflecting the absence of significant genotoxic stress, as observed in a number of non-AD fibroblasts and in radioresistant controls. (2) Cytoplasmic pATM broad staining with nuclear pATM foci, reflecting the presence of significant genotoxic stress, as observed in some cases of non-AD fibroblasts showing spontaneous DSB. (3) Perinuclear pATM crowns with nuclear pATM foci, suggesting that pATM molecules were still able to cross the nuclear membrane, as observed in AD fibroblasts only. (4) Perinuclear pATM crowns without pATM foci, suggesting that pATM molecules were unable to cross the nuclear membrane, as observed in AD fibroblasts only. The ordered succession of these four patterns suggested a specific response to stress that may result in an agglutination or a “traffic-jam” of ATM molecules around the nuclear membrane, preventing the ATM nucleoshuttling and maybe the diffusion of ZOPRA molecules (Figure 6D).

After irradiation, the number of these perinuclear pATM crowns (patterns 3 and 4) drastically decreased 10 min after 2 Gy, regardless of the AD fibroblasts tested (Figure 6E). Since the pATM signals disappeared after irradiation, these findings suggest that the irradiation triggers the monomerization of ATM dimers, at least transiently.

### 3.4. AD Fibroblasts Show a Specific Protein Partnership between ATM and APOE Proteins

In the frame of the RIANS model, the ATM dimers dissociated in monomers in response to the oxidative stress [17], as already suggested [32]. Such monomers may be sequestrated in the cytoplasm by overexpressed protein ATM substrates, called “X-proteins”: we have shown that each genetic disease associated with moderate radiosensitivity and delayed RIANS was characterized by one X-protein, at least [17]. We therefore examined which X-protein may be specifically both overexpressed in the cytoplasm and phosphorylated by ATM in AD fibroblasts. Among the potential X-protein candidates for AD, we quickly eliminated the β-amyloid peptides and the tau proteins since they did not show evident perinuclear staining in the 10 AD fibroblasts tested. In contrast, it appeared that the APOE protein, that is currently documented in AD research [43], showed: (1) Spontaneous cytoplasmic and perinuclear localization of APOE protein in the 10 AD fibroblasts tested, similar to fibroblasts provided from some other neurodegenerative or aging syndromes, when observed with anti-*APOE* immunofluorescence (Figure 7A). (2) Spontaneous cytoplasmic overexpression of APOE and pATM proteins, higher in all the AD fibroblasts tested than in controls, when observed by immunoblots with cytoplasmic extracts (Figure 7B). (3) A specific cytoplasmic interaction between ATM and APOE proteins in all the AD fibroblasts tested when observed with the proximity ligation assay (PLA) (Figure 7C).

Particularly, the PLA revealed dots (or foci) of interaction between ATM and APOE proteins that were found significantly more numerous in the cytoplasm of AD cells than in radioresistant controls. In fibroblasts derived from neurodegenerative or aging syndromes such as Werner’s syndrome, the PLA foci were either absent or much less numerous (*p* < 0.001). Lastly, it is noteworthy that such PLA dots were also absent in *ATM*-mutated fibroblasts. It is also noteworthy that there is no human syndrome associated with null expression of APOE to serve as a negative control (Figure 7C).

Interestingly, the in silico analysis of the *APOE* sequence revealed the presence of two putative ATM phosphorylation SQ/TQ domains defined elsewhere [44]: the APOE^ser72^ and the APOE^thr75^. In order to investigate whether these domains were involved in the interaction between ATM and APOE, we developed phosphospecific antibodies against both phosphorylated forms. Immunofluorescence analysis clearly showed that both spontaneous pAPOE signals were mainly localized in the cytoplasm and were stronger in the cytoplasm than in the nucleus. Furthermore, the pAPOE signals were also higher in AD fibroblasts than in controls. However, the conclusions reached with immunoblots were less clear: while both anti-*pAPOE^ser72^* and anti-*pAPOE^thr75^* antibodies revealed significant cytoplasmic forms of pAPOE, the anti-*pAPOE^thr75^* antibody did not provide reproducible and clear signals from nuclear extracts, suggesting a potential rarity and/or instability of the specific *pAPOE^thr75^* in the nucleus (Figure 7D,E).

Altogether, these data were consistent with the formation of perinuclear pATM crowns in non-irradiated AD fibroblasts that may contain specific ATM-phosphorylated forms of APOE protein.

## 4. Discussion

The initial goal of this study was to document, at the molecular level, the radiosensitivity of AD fibroblasts that was previously pointed out at the cellular scale [5,6,7]. To this aim, we deliberately chose to apply a routine approach based on the RIANS model. Such approach has been successfully applied to a number of genetic diseases, including aging syndromes [22,23,24,25,26,27]. During our investigations, a specific perinuclear localization of the pATM forms was noticed in all the non-irradiated AD fibroblast cell lines tested. This feature was not observed in any other fibroblast cell lines derived from the aging, cancer syndromes or apparently healthy donors tested at the same cell culture passages. In addition, the percentage of cells with perinuclear pATM crowns increased with the cell culture passage in AD fibroblasts, suggesting an age-dependent evolution. Such perinuclear pATM crowns disappeared, transiently at least, after an irradiation. We therefore hypothesized that these perinuclear crowns might be used as a specific biomarker of AD, for the early detection of AD, the progression of the disease, and potentially, for its treatment.

As described in Section 1, in the frame of the RIANS model, the RI oxidative stress causes the formation of ATM monomers from the cytoplasmic ATM dimers. The ATM monomers diffuse in the nucleus and participate to the recognition of DSB by NHEJ. A delayed RIANS caused by the sequestration of ATM monomers in the cytoplasm via overexpressed cytoplasmic ATM substrates (called X-proteins) provides moderate radiosensitivity associated with cancer proneness or accelerated aging [17]. Here, four molecular endpoints (micronuclei, γH2AX, pATM and MRE11) converged to the same conclusions. Altogether, our data, through the RIANS model, provide the first molecular explanation of the radiosensitivity of cells derived from AD patients, in agreement with the historical observations at the cellular scale [5,6,7].

The spontaneous perinuclear localization of pATM forms prompted us to identify any X-protein candidate that would localize very close to the nuclear membrane in AD cells. Unlike β-amyloid peptides and tau proteins, the APOE protein reached the three major requirements: (1) APOE was found implicated in certain forms of AD, although there is still confusion between the roles of an overexpressed APOE protein in cells and the *APOE*ε polymorphism status [43]. (2) In silico studies showed that APOE holds putative ATM phosphorylation SQ/TQ domains [44]. (3) APOE, as an apolipoprotein, was found to be involved in the protein trafficking through cellular and nuclear membranes and the regulation to cholesterol [45,46,47]. Four observations consolidated our hypothesis: (1) APOE was found spontaneously more expressed in all the AD fibroblasts tested than in the controls (Figure 8B), (2) a partnership between ATM and APOE may occur through SQ/TQ phosphorylation (Figure 8), (3) PLA data suggested that some ATM-APOE complexes may be found close to the nuclear membrane (Figure 7C), and (4) the localization of the phosphospecific forms of the APOE (pAPOE) was consistent with the ATM–pAPOE interaction (Figure 7C,D). At this point, it is important to recall that the anti-*pATM^ser1981^* antibodies specifically reveal auto-phosphorylated dimers and not ATM monomers [17]. Hence, the presence of a perinuclear pATM crown revealed by anti-*pATM* immunofluorescence suggests a high concentration of ATM dimers close to the nuclear membrane.

What scenario may unify both the high accumulation of ATM dimers and ATM-pAPOE complexes? A permanent oxidative stress is responsible for the formation of spontaneous DNA breaks, but also for a continuous monomerization of the ATM dimers, as already suggested by treating cells with hydrogen peroxide [32]. Hence, some ATM monomers may be sequestrated in the close vicinity of the nuclear membrane by APOE proteins, explaining the specific phosphorylated forms of APOE (pAPOE) revealed by the pAPOE^ser72^ or pAPOE^thr75^ immunofluorescence and the PLA data (Figure 7). Once the ATM-pAPOE complexes form, the stress-induced diffusion of ATM monomers may progressively cause a highly localized concentration of ATM monomers, leading to ATM re-dimerization, and therefore producing a pATM perinuclear crown, easily visible with anti-*pATM* immunofluorescence. Hence, this model of accretion of ATM monomers strongly suggests that the perinuclear crowns of pATM are composed of an inner layer formed by ATM monomers, and that pAPOE forms and an outer layer formed by ATM dimers. In a cumulative and progressive process, this phenomenon may specifically contribute in AD cells to an increase of the number of cells with perinuclear pATM crowns and an increase of the crown “thickness”. Such model is, therefore, consistent with the patterns 1, 2, 3 and 4 described in Figure 8.

In order to support the above mechanistic model, and notably the occurrence of the different patterns of the pATM perinuclear crowns shown in Figure 3, we designed a brand-new mathematical approach consisting of six nonlinear differential equations. Each equation describes the kinetics of the concentrations of a specific protein form involved in the mechanistic model: D_C_ and M_C_, time-dependent functions, reflect the ATM dimers and the ATM monomers in the cytoplasm, respectively. D_A_, A and A_M_, time-dependent functions, reflect the dimers in the immediate vicinity of the nucleus, the free APOE monomers and the ATM-pAPOE complexes, respectively (Figure 8). Lastly, M_N_ reflects the nuclear ATM monomers varying with time. This mathematical approach was applied to simulate the consequences of a continuous oxidative stress induction with a low (such as in the controls) or a high cytoplasmic APOE protein level (such as in AD cells) (Figure 8). The resulting blue kinetics reflecting the control data clearly showed a slow production of ATM dimers, the absence of ATM-pAPOE complexation and a facilitated diffusion of ATM monomers in the nucleus (Figure 8). In contrast, in AD cells, the ATM-pAPOE complexation increased together with the formation of ATM dimers in the close vicinity of the nucleus. As a result, the function reflecting the ATM monomers in the nucleus rapidly decreased to zero. By further investigating this mathematical modeling, it was possible to explain the four perinuclear pATM crown patterns defined above. To this aim, initial A(0) parameter values were modified.

By considering both biological and mathematical models, the fact that the ZOPRA treatment did not fully correct the molecular AD phenotype with all four molecular endpoints tested (micronuclei, γH2AX, pATM and MRE11) may be supported by the high concentration of ATM proteins cumulated in the close vicinity of the nucleus that did not help the statins and bisphosphonates to diffuse in the nucleus. In contrast, both models can also explain that a dose of IR, such as that mimicking a standard radiotherapy session, may cause a massive monomerization of ATM dimers early after irradiation, leading to the destruction of the perinuclear pATM crowns. Further investigations are, however, required to evaluate the benefit of a ZOPRA treatment in AD patients after radiotherapy, and to explore whether radiotherapy may influence the pattern and/or the number of perinuclear pATM crowns, as suggested in Figure 6D.

The spontaneous perinuclear pATM and pAPOE crowns found specifically in AD fibroblasts raised several questions about their potential role as specific biomarkers of AD: can they help in the detection of AD, the estimation of the disease progression and/or the definition of a possible predisposition to AD? Our experimental findings and the associated mathematical model were consistent with a progressive acceleration of aging due to non-recognized DNA breaks that cumulate in the nucleus throughout life. Such statement is consistent with numerous studies suggesting that AD results from a cumulative oxidative stress [48,49]. Can our observations in AD fibroblasts reflect the same biological features in brain cells, notably in hippocampus cells? The relevance of the RIANS model has been verified in numerous tissues, including brain astrocytes [50]. Furthermore, cutaneous fibroblasts were shown to be the tissue model more representative of the individual response to IR than the other tissues [8]. Nevertheless, to sample brain cells would raise many more ethical and practical problems than skin biopsies. While brain cells are characterized by a low concentration of ATM protein [51] and most ATM molecules were found cytoplasmic [28,29,30,31,32]. our findings may be consistent with a slower and less efficient ATM nucleoshuttling in brain cells, accompanied by a higher sensitivity to DNA breaks, with more significant clinical consequences than those observed in skin fibroblasts. Further investigations are needed to verify this hypothesis.

The cumulative nature of the perinuclear pATM and pAPOE crowns’ formation also raised the question of a correlation between the yield or the pattern of the crowns and the disease progression. In this study, the clinical data about the progression of the disease in the 10 AD cell lines were not available. Besides, it must be stressed that post-mortem human brain data would not be useful to answer this question since they would not provide the threshold of perinuclear pATM crowns necessary for significant clinical features. Furthermore, the causes of cellular death and the progression of AD are not likely to be equally correlated with the perinuclear pATM crowns. Again, further clinical investigations will help document the AD progression and its associated clinical criteria.

Finally, our findings raised the question of a potential predisposition to AD that would be based on the existence of a significant amount of perinuclear pATM crowns at the earliest age. However, our experimental data and the mathematical model suggest that the environmental stress and the genetic factors may unequally contribute to increase or even accelerate the formation of perinuclear pATM crowns. Such hypothesis may explain the heterogeneity of the AD cases, notably vis-à-vis the age at diagnosis.

## 5. Conclusions

Altogether, our experimental findings and the mathematical model were consistent with the hypothesis that in the development of AD, there is a progressive accumulation of ATM-pAPOE and pATM-pATM dimeric complexes around the nuclear membrane in response to oxidative stress throughout life, facilitated by the predisposition of overexpressed APOE protein in the cytoplasm. Such specific subcellular localization of ATM may prevent the RIANS, i.e., the recognition and repair of DNA strand breaks by the ATM kinase. Irradiation may make the perinuclear pATM crowns disappear. For the first time, to our knowledge, the ATM kinase was shown to be involved in the pathology of AD, and its interaction with APOE may play a major role in aging, which may lead to a new approach in the understanding of AD. Particularly, with further investigations, our findings may help in the definition and the detection of individual predisposition to AD, helpful in the early diagnosis of AD.

## Figures and Tables

**Figure 1 cells-12-01747-f001:**
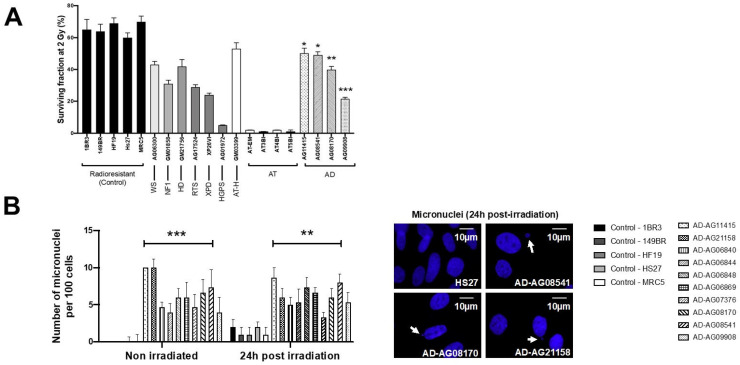
Cellular radiosensitivity and cytogenetic features of AD fibroblasts. (**A**) Surviving fraction at 2 Gy (SF2) of the indicated fibroblast cell lines from the indicated syndromes (WS: Werner’s syndrome; NF1: neurofibromatosis type 1; HD: Huntington’s disease; RTS: Rothmund–Thomson syndrome; XPD: Xeroderma Pigmentosum D; HGPS: Hutchinson–Gilford progeroid syndrome; ATH: ataxia telangiectasia heterozygous; AT ataxia telangiectasia; AD: Alzheimer’s disease). Each bar represents the mean ± standard error of the mean (SEM) of three replicates, at least. The asterisks are shown for AD cells vs. controls only. (**B**) Representative immunofluorescence images of micronuclei observed 24 h post-irradiation (2 Gy X-rays) in the indicated cells with DAPI staining. The arrows show isolated micronuclei during exonucleosis. Histogram showing the number of micronuclei per 100 cells assessed before irradiation or 24 h post-irradiation. Each bar represents the mean ± SEM of three replicates, at least. **, *p* < 0.01 and ***, *p* < 0.001.

**Figure 2 cells-12-01747-f002:**
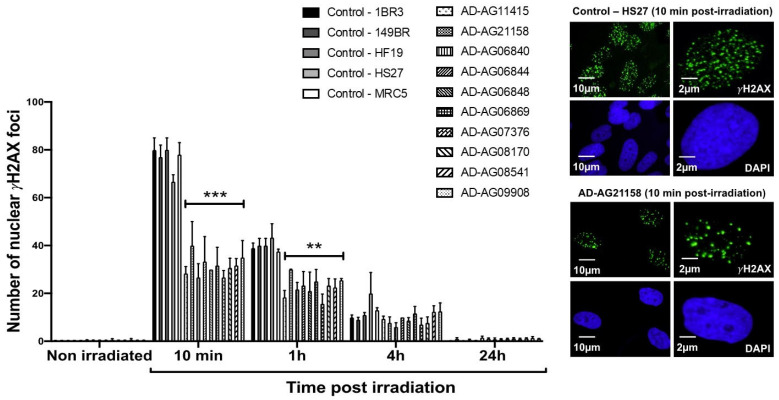
γH2AX foci data from AD fibroblasts. Representative immunofluorescence images of nuclear γH2AX foci observed 10 min after irradiation (2 Gy X-rays) in the indicated cells. The DAPI signal served as nuclear counterstaining. Histogram showing the number of nuclear γH2AX foci per cell assessed before irradiation or at 10 min, 1 h, 4 h and 24 h post-irradiation. Each bar represents the mean ± SEM of three replicates, at least. **, *p* < 0.01 and ***, *p* < 0.001.

**Figure 3 cells-12-01747-f003:**
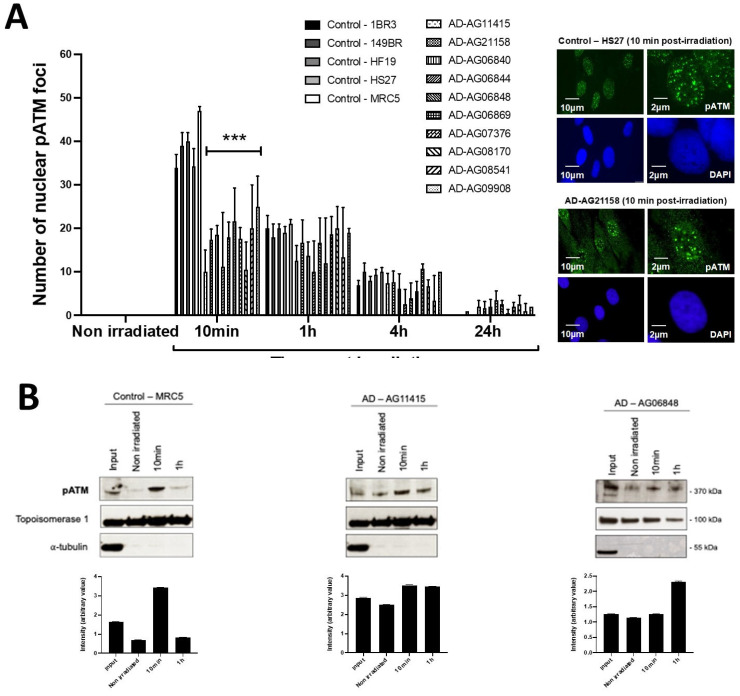
pATM foci and immunoblot data from AD fibroblasts. (**A**) Representative immunofluorescence images of nuclear pATM foci observed 10 min after irradiation (2 Gy X-rays) in the indicated cells. The DAPI signal served as nuclear counterstaining. Histogram showing the number of nuclear pATM foci per cell assessed before irradiation or at 10 min, 1, 4 and 24 h post-irradiation. Each bar represents the mean ± SEM of three replicates, at least. ***, *p* < 0.001. (**B**) Immunoblots performed with anti-*pATM* antibodies in nuclear extracts from cells processed before irradiation or at 10 min and 1 h post-irradiation (2 Gy X-rays). Topoisomerase I and a-tubulin served as cell fractionation and loading controls. The corresponding quantified Western blot bands were expressed as arbitrary units.

**Figure 4 cells-12-01747-f004:**
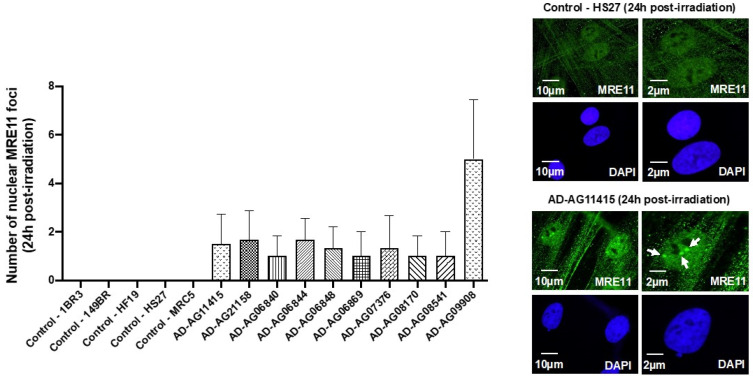
MRE11 foci from AD fibroblasts. Representative immunofluorescence images of nuclear MRE11 foci observed 24 h after irradiation (2 Gy X-rays) in the indicated cells. The DAPI signal served as nuclear counterstaining. Histogram showing the number of nuclear MRE11 foci per cell assessed before irradiation or at 24 h post-irradiation. Each bar represents the mean ± SEM of three replicates, at least. White arrows indicate the nuclear MRE11 foci.

**Figure 5 cells-12-01747-f005:**
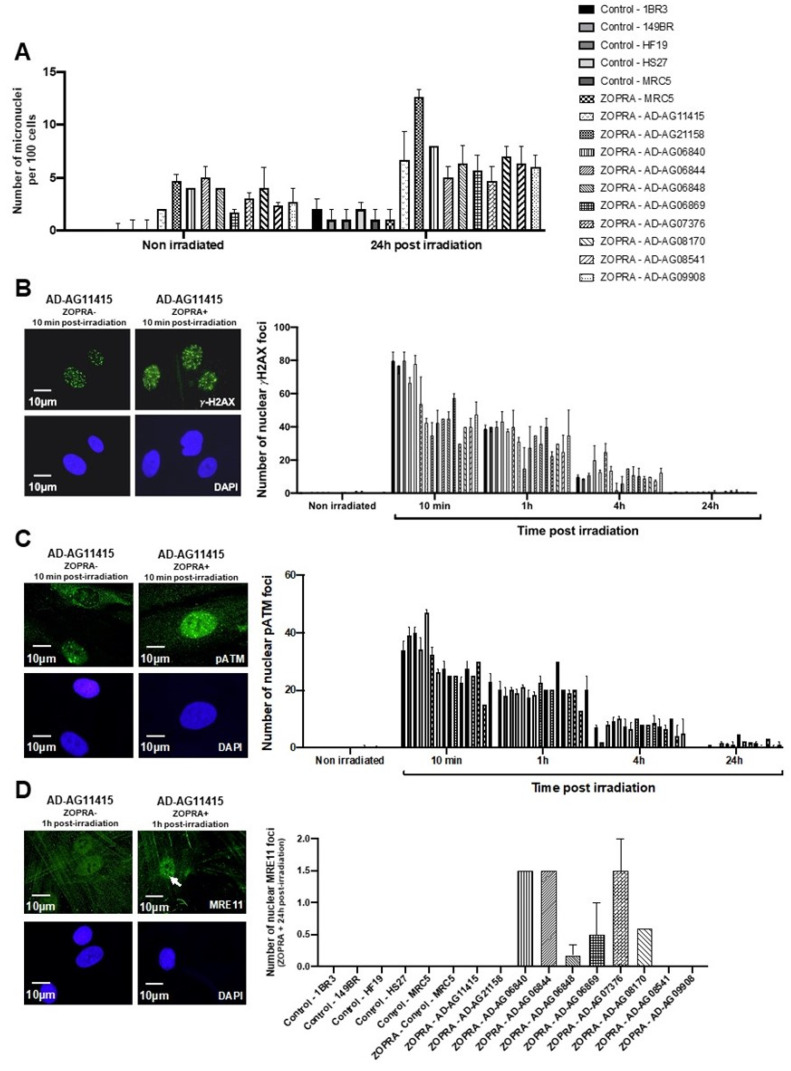
Effects of the ZOPRA treatment on AD fibroblasts. (**A**) Histogram showing the number of micronuclei per 100 cells in ZOPRA- and non-ZOPRA-treated cells, as indicated. After ZOPRA treatment, cells were exposed at 2 Gy X-rays, followed by 24 h for repair. The non-ZOPRA-treated cell data are those shown in Figure 1A. Each bar represents the mean ± standard error of the mean (SEM) of three replicates, at least. (**B**) Representative immunofluorescence images of nuclear γH2AX foci observed after or without ZOPRA treatment, followed by 2 Gy X-rays and 10 min for repair. The DAPI signal served as nuclear counterstaining. Histogram showing the number of nuclear γH2AX foci per cell observed in ZOPRA-treated cells irradiated at 2 Gy, followed by the indicated times for repair. Each bar represents the mean ± SEM of three replicates, at least. (**C**) Representative immunofluorescence images of nuclear pATM foci observed after or without ZOPRA treatment, followed by 2 Gy X-rays and 10 min for repair. The DAPI signal served as nuclear counterstaining. Histogram showing the number of nuclear pATM foci per cell observed in ZOPRA-treated cells irradiated at 2 Gy, followed by the indicated times for repair. Each bar represents the mean ± SEM of three replicates, at least. (**D**) Representative immunofluorescence images of nuclear MRE11 foci observed after or without ZOPRA treatment, followed by 2 Gy X-rays and 24 h for repair. The DAPI signal served as nuclear counterstaining. Histogram showing the number of nuclear pATM foci per cell observed in ZOPRA-treated cells irradiated at 2 Gy, followed by 24 h for repair. Each bar represents the mean ± SEM of three replicates, at least. White arrows indicate nuclear pATM foci.

**Figure 6 cells-12-01747-f006:**
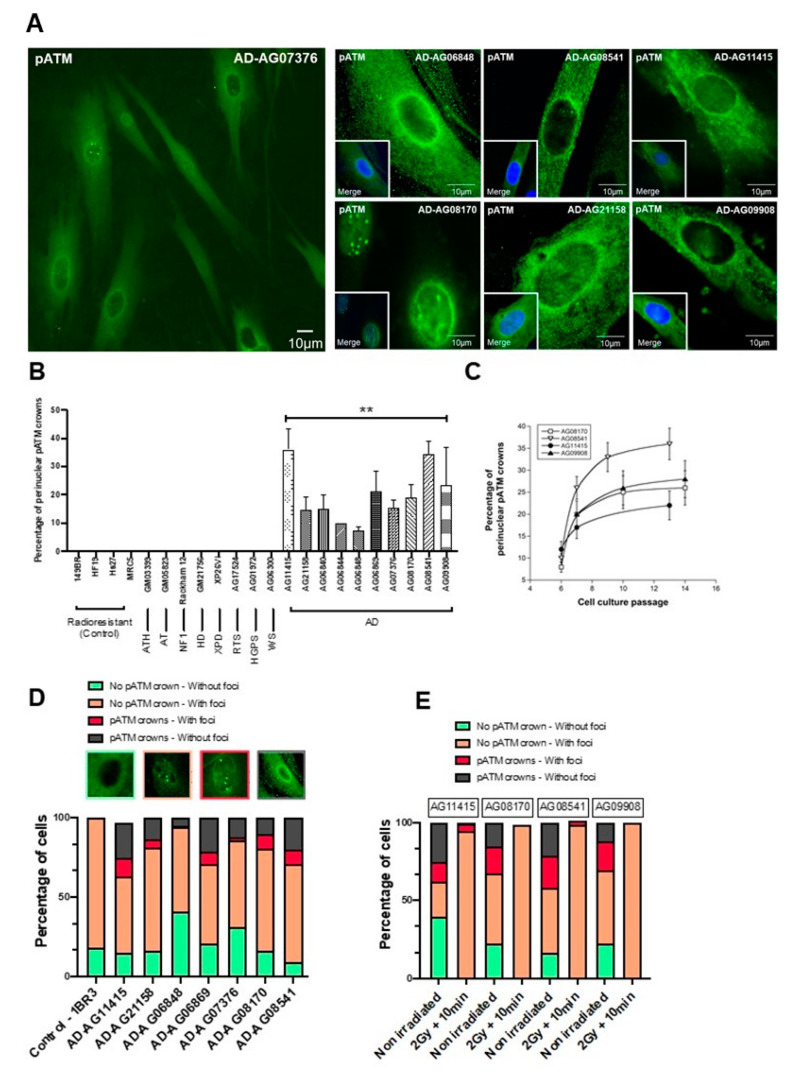
Specific spontaneous perinuclear localization of pATM in AD fibroblasts. (**A**) Representative immunofluorescence images of the perinuclear localization of pATM protein in AD cells. The DAPI signal served as nuclear counterstaining. (**B**) Histogram showing the percentage of spontaneous perinuclear pATM crowns observed in the indicated non-irradiated cells after an anti-*pATM* immunofluorescence. Each bar represents the mean ± SEM of four replicates, at least. **, *p* < 0.01. (**C**) Percentage of perinuclear pATM crowns as a function of cell culture passage in the indicated AD cells. Each point represents the mean ± SEM of three replicates. (**D**) Relative proportions of four specific patterns of the perinuclear pATM crowns observed in the indicated cells after anti-*pATM* immunofluorescence. The data were obtained during the experiments that served for the quantification of the pATM crowns in cells (shown in panel **B**). (**E**) Relative proportions of four specific patterns of the perinuclear pATM crowns observed in the indicated cells after anti-*pATM* immunofluorescence, assessed before or after irradiation (2 Gy + 10 min).

**Figure 7 cells-12-01747-f007:**
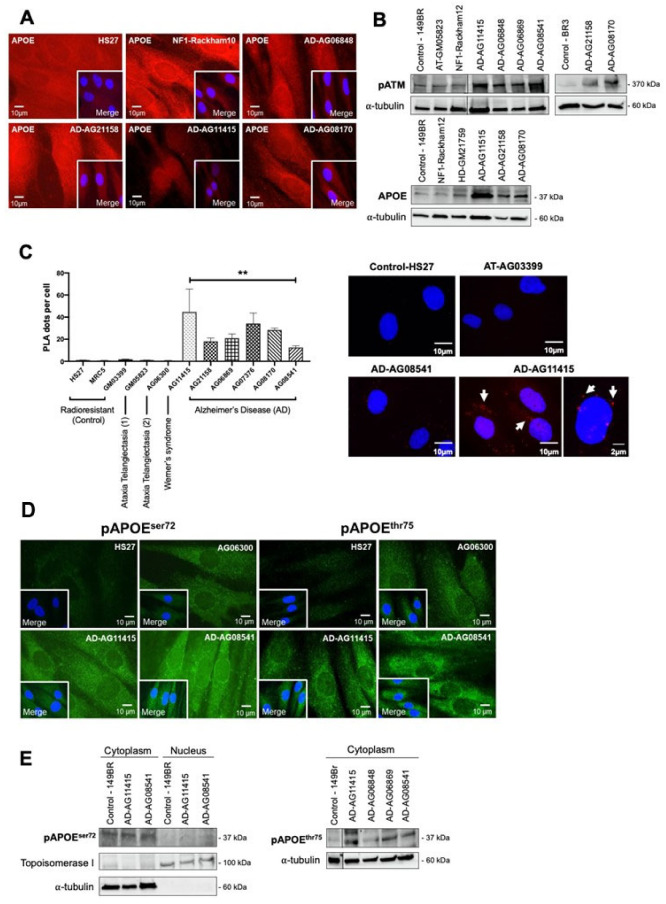
ATM and APOE interactions in AD fibroblasts. (**A**) Representative immunofluorescence images of the spontaneous APOE subcellular localization in the indicated non-irradiated cells. The DAPI signal served as nuclear counterstaining. (**B**) Immunoblots performed with anti-*pATM* and anti-*APOE* antibodies on cytoplasmic extracts from the indicated non-irradiated cells. Alpha-tubulin served as a cell fractionation and loading control. (**C**) The proximity ligation assay (PLA) with anti-*ATM* and anti-*APOE* antibodies was performed on the indicated non-irradiated cells. Histogram showing the PLA dots scored by immunofluorescence microscopy, that represent the detected ATM–APOE interaction. Each plot represents the mean ± SEM of three replicates. Representative PLA immunofluorescence images in the indicated non-irradiated cells. The DAPI signal served as nuclear counterstaining. **, *p* < 0.01. (**D**) Representative immunofluorescence images of the spontaneous pAPOE^ser72^ and pAPOE^thr75^ subcellular localization in the indicated non-irradiated cells. The DAPI signal served as nuclear counterstaining. (**E**) Immunoblots performed with anti-*pAPOE^ser72^* and -*pAPOE^thr75^* antibodies in cytoplasmic or nuclear extracts from the indicated cells. Topoisomerase I and alpha-tubulin served as loading controls and as cytoplasmic and nuclear cell fractionation controls, respectively.

**Figure 8 cells-12-01747-f008:**
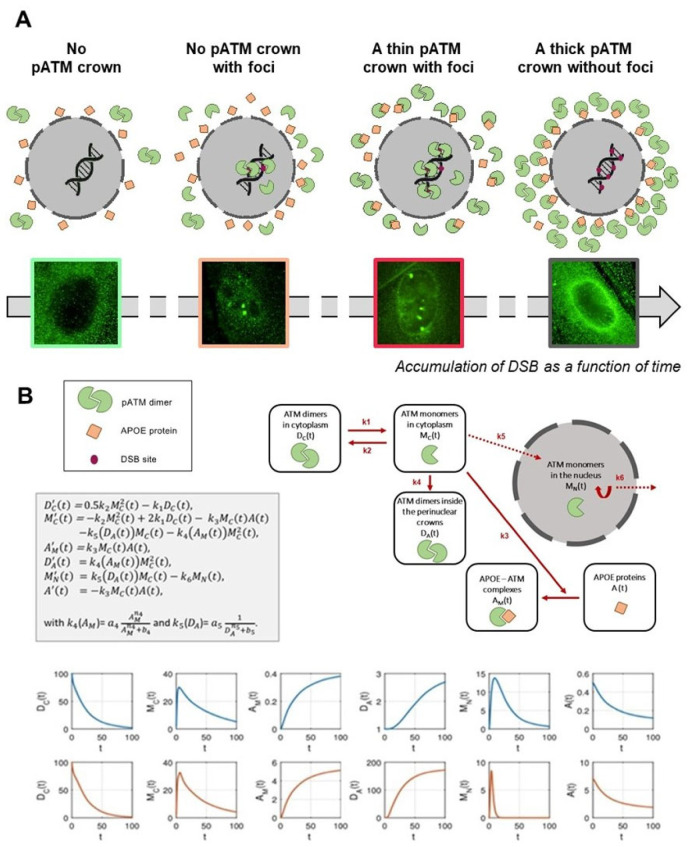
(**A**) Mechanistic model of the formation of the perinuclear pATM crowns in AD fibroblasts. The environmental stress, amplified by the potential predisposition to DNA damage impairment, produces oxidative stress, causing DNA strand breaks and monomerization of cytoplasmic ATM dimers, consistent with the RIANS model. In a context in which APOE, an ATM phosphorylation substrate, is overexpressed, the ATM monomers phosphorylate and preferentially bind to APOE by forming ATM-APOE complexes, instead of diffusing in the nucleus to recognized DSB, and trigger their repair by NHEJ. These complexes progressively prevent the ATM nucleoshuttling, which increases the concentration of ATM monomers around the nucleus. As a result, ATM monomers re-associate by forming ATM dimers around the nucleus, which prevents recognition of DSB that accumulate in nucleus. This re-dimerization of ATM monomers is easily visible by immunofluorescence as perinuclear pATM crowns. (**B**) Mathematical modeling of the formation of perinuclear pATM crowns. The first equation in Dc represents the variation of ATM dimers, Dc’(t). The first term on the right side represents the formation of a dimer from two monomers, and the second term stands for the monomerization of an ATM dimer into two ATM monomers. Similarly, in the second equation, the right side describes, respectively, the dimerization, the monomerization, the APOE-ATM complex formation, the passage to the nucleus and the pATM crown, populated by dimerization in the crown. The third equation describes the formation of the APOE-ATM complex, and the fourth equation describes the crowding of the crown through dimerization of ATM monomers in the nucleus neighborhood. The fifth equation represents the passage to the nucleus with a degradation term, and the last equation stands for the loss of the free APOE protein to form a complex with ATM monomers. Representative examples of the variations of each parameter are shown in blue for non-AD radioresistant controls and in red for AD cells.

**Table 1 cells-12-01747-t001:** Major characteristics of the fibroblast cell lines used in this study.

Cell Lines	Gender	Associated Disease	Radiation Response	Age (at Sampling)	APOE Genotype	Origin
1BR3	Male	Apparently healthy	Radioresistant	3	nd	ECACC
149BR	Male	Apparently healthy	Radioresistant	51	nd	ECACC
HF19	Female	Apparently healthy	Radioresistant	fetus	nd	ECACC
Hs27	Male	Apparently healthy	Radioresistant	<1 month	ε3/ε4	ATCC
MRC5	Male	Apparently healthy	Radioresistant	children	ε3/ε4	ECACC
GM03399	Female	AT heterozygous mutation	Radioresistant	20	nd	Coriell Institute
AT2EM	nd	nd	Hyper-radiosensitive	nd	nd	COPERNIC
AT3BI	Male	nd	Hyper-radiosensitive	4	nd	COPERNIC
AT4BI	Male	nd	Hyper-radiosensitive	6	nd	COPERNIC
GM05823 (AT5BI)	Male	AT Compound heterozygous mutation	Hyper-radiosensitive	18	nd	Coriell Institute
Rackham 12	Male	Neurofibromatosis type I	Moderately radiosensitive	23	nd	COPERNIC
GM21756	Female	Huntington’s disease	Moderately radiosensitive	nd	nd	Coriell Institute
XP26VI	nd	Xeroderma Pigmentosum D	Moderately radiosensitive	15	nd	COPERNIC
AG17524	Female	Rothmund–Thomson Syndrome	Moderately radiosensitive	4	nd	Coriell Institute
AG01972	Female	Hutchinson–Gilford syndrome	Hyper-radiosensitive	14	nd	Coriell Institute
AG06300	Male	Werner’s syndrome	Hyper-radiosensitive	37	nd	Coriell Institute
AG11415	Male	AD	Moderately Radiosensitive	55	ε3/ε3	Coriell Institute
AG21158	Female	AD	Moderately Radiosensitive	69	ε2/ε3	Coriell Institute
AG06840	Male	AD *PSEN1* mutation	Moderately Radiosensitive	56	ε3/ε3	Coriell Institute
AG06844	Male	AD Family history early onset	Moderately Radiosensitive	59	ε3/ε4	Coriell Institute
AG06848	Female	AD *PSEN1* mutation	Moderately Radiosensitive	56	ε3/ε4	Coriell Institute
AG06869	Female	AD	Moderately Radiosensitive	60	ε4/ε4	Coriell Institute
AG07376	Male	AD	Moderately Radiosensitive	60	ε3/ε3	Coriell Institute
AG08170	Male	AD Family history early onset	Moderately Radiosensitive	56	ε3/ε4	Coriell Institute
AG08541	Female	AD	Moderately Radiosensitive	59	ε3/ε3	Coriell Institute
AG09908	Female	AD *PSEN2* mutation	Moderately Radiosensitive	81	ε3/ε3	Coriell Institute

Nd: not determined; ECACC: European Collection of Authenticated Cell Cultures; ATCC: American Type Culture Collection.

## Data Availability

The data presented here are either present in a deposited database (see the Materials and Methods Section) or will be made available upon reasonable request.

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
