# Peer review of "Toward an Early Diagnosis for Alzheimer’s Disease Based on the Perinuclear Localization of the ATM Protein"

_cells, 2023, doi:10.3390/cells12131747_

Round 1

Reviewer 1 Report

Minor grammatical errors need to be corrected in the manuscript. 

Author Response

REVIEWER 1 :

We thank the reviewer for his/her comments.

The manuscript "Toward an early diagnosis for Alzheimer’s disease based on the perinuclear localization of the ATM protein" submitted makes a significant contribution to research wherein the authors observed a perinuclear localization of the ATM protein specifically in the Alzheimer’s disease (AD) fibroblasts and further proposed a mechanistic interpretation of AD deriving from the RIANS model. Their findings suggest that pATM and/or pAPOE may serve as biomarkers for an early reliable diagnosis of AD.

Comments/Suggestions to Authors:

  1. Authors are advised (if the keyword limit permits) to include “DSB” and “Irradiation” in the ‘keywords’ section to improve visibility of their article.

OK see modified text in the keywords.

  1. Abstract is well-written, but it's crucial to bridge or explicitly address the gap. Please include a brief sentence in the abstract that “clearly” describes the research gap.

OK see modified text in the abstract.

  1. Authors did a good job by providing adequate information in Background/Introduction section.

OK thank you for this comment

  1. Have the authors initially tested the effects of Pravastatin and Zoledronate using MTT assay on the fibroblast cell lines used in this study?

No for two reasons. First, because the MTT assay, by comparison with the cell survival assay may underestimate survival: senescence is linked to an irreversible G0/G1 arrest that may appear as positive signal in MTT and negative in clonogenic cell survival. Second, whatever MTT or clonogenic cell survival, to test ZOPRA treatment with cell death assays is complex since we have to manage longer times than with molecular assays and a long list of control is necessary. Since we have described the link between clonogenic cell survival and molecular endpoints (see Le Reun et al. IJMS, 2022), we have deliberately chosen to test ZOPRA with molecular assays only.

  1. Please provide a justification to use Pravastatin (24h) before and then Zoledronate (12h) for the combination treatment?                                    Such protocol is due to the different life periods of the drugs (varela et al., Nature Med 2008) and the fact that we have applied the same protocols for more than ten papers already (see restier-verlet et al. Biomolecules 2023, Vol 13). See modified text in materials and methods.

  1. Is cell line authentication performed, and if so, what method is used?          Cell culture quality control is insured by commercial repositories. With regard the cell lines isolated from the lab, the respect of GPL is including in the declaration of the collection to the French Ministry of Research. See modified text in materials and methods.

  1. Minor grammatical errors in the manuscript should be corrected.             The text has been edited

  1. In the conclusion section, write a few sentences about the future perspectives of this research.

OK see modified text in the conclusion

Reviewer 2 Report

The article “Toward an early diagnosis for Alzheimer’s disease based on the perinuclear localization of the ATM protein” investigates the effect of pATM forms and/or pAPOE as biomarkers for a reliable early diagnosis of Alzheimer’s disease (AD) on fibroblast samples.

Overall, the article is scientifically sound, well-written, coherent, understood by non-experts, and represents cutting-edge research.

The experimental methodology (based on eleven untransformed AD fibroblasts & non-AD fibroblasts from cancer or neurodegenerative diseases and five from healthy samples) justifies the results. Also, the results are interpreted by a biomathematical model based on nonlinear differential equations that supports the mechanistic model by describing the kinetics of the concentrations of an ATM protein involved in the mechanistic model. Thus, the conclusion that ATM kinase is involved in AD pathology and could play a significant role in aging relies on a thoroughly systematic methodology and might open a new approach to understanding AD disease.

Minor issues are: defining the abbreviation “ATM” and spelling mistakes (e.g., “from cells cells collected” line 130)

The article can be published in its present form.

Author Response

REVIEWER 2 :

We thank the reviewer for his/her comments.

The article “Toward an early diagnosis for Alzheimer’s disease based on the perinuclear localization of the ATM protein” investigates the effect of pATM forms and/or pAPOE as biomarkers for a reliable early diagnosis of Alzheimer’s disease (AD) on fibroblast samples.

Overall, the article is scientifically sound, well-written, coherent, understood by non-experts, and represents cutting-edge research.

The experimental methodology (based on eleven untransformed AD fibroblasts & non-AD fibroblasts from cancer or neurodegenerative diseases and five from healthy samples) justifies the results. Also, the results are interpreted by a biomathematical model based on nonlinear differential equations that supports the mechanistic model by describing the kinetics of the concentrations of an ATM protein involved in the mechanistic model. Thus, the conclusion that ATM kinase is involved in AD pathology and could play a significant role in aging relies on a thoroughly systematic methodology and might open a new approach to understanding AD disease.

Minor issues are: defining the abbreviation “ATM” and spelling mistakes (e.g., “from cells cells collected” line 130)

OK see modified text

The article can be published in its present form.

Reviewer 3 Report

In this manuscript, the authors explore the association between the ATM pathway and Alzheimer’s disease via applying an approach based on the RIANS model in 10 AD fibroblasts cell lines. The results showed that spontaneous perinuclear localization of p-ATM is important to the AD process. The authors present lots of interesting results. In general, my assessment of this manuscript is positive, although I believe that it needs to be further improved prior to publication in Cells.

Specific comments:

1.       In all figure’s legends, the author did not present how many cells were quantified?

2.       In Figure 3B, I suggest quantifying the western blot bands of p-ATM via 3 repeats.

3.       In Figure 7B and 7E, I suggest the author also test the total ATM or APOE protein expression.

Author Response

REVIEWER 3 :

We thank the reviewer for his/her comments.

In this manuscript, the authors explore the association between the ATM pathway and Alzheimer’s disease via applying an approach based on the RIANS model in 10 AD fibroblasts cell lines. The results showed that spontaneous perinuclear localization of p-ATM is important to the AD process. The authors present lots of interesting results. In general, my assessment of this manuscript is positive, although I believe that it needs to be further improved prior to publication in Cells.

Specific comments:

  1. In all figure’s legends, the author did not present how many cells were quantified?

 For clarity and since we applied routine protocols with different techniques, we have deliberately chosen to indicate the number of cells tested for each technique in Materials and methods. See modified text.

  1. In Figure 3B, I suggest quantifying the western blot bands of p-ATM via 3 repeats.                                                                                                        See modified legend and figure 3.

  1. In Figure 7B and 7E, I suggest the author also test the total ATM or APOE protein expression.

 This sentence may have two different meanings:

  • If we consider “total” extracts: this section is clearly devoted to the cytoplasmic expression of the X-protein and ATM forms and may we recall that the total extracts with ATM is a mixture of ATM dimers and monomers and, with regard to APOE, again, the most important requirement is to know whether APOE is abundant in cytoplasm (which may be independent of the abundancy in nucleus)
  • If we consider ‘total’ protein : we provided total APOE in both immunoblot and immunofluorescence in Fig 7B. In both cases, the localization of APOE appears clearly cytoplasmic. With regard to ATM, as said above, anti-ATM antibody does not discriminate ATM dimers and monomers. However, line 305, it is written, “qPCR analysis of ATM synthesis did not reveal any difference between radioresistant controls and AD fibroblasts”. Furthermore, we provide you here an example of a Western Blot with anti-ATM antibody reaching the same conclusions (Figure A, see below).
  • In addition, we have deeply modified the text related to the figure 7E to stress again on the structure of the perinuclear pATM crown from data obtained with phosphospecific pAPOE antibodies. See modified text .

Figure A : Immunoblots with ATM and Ku80 antibody from total extracts of control MRC5 and the 2 indicated AD cells

Round 2

Reviewer 3 Report

Aceept